# Reversible metal cluster formation on Nitrogen-doped carbon controlling electrocatalyst particle size with subnanometer accuracy

Janis Timoshenko [1] ✉, Clara Rettenmaier [1], Dorottya Hursán [1], Martina Rüscher[1], Eduardo Ortega [1], Antonia Herzog [1], Timon Wagner [1], Arno Bergmann [1], Uta Hejral [1], Aram Yoon[1], Andrea Martini[1], Eric Liberra[1], Mariana Cecilio de Oliveira Monteiro [1] & Beatriz Roldan Cuenya [1] ✉

Copper and nitrogen co-doped carbon catalysts exhibit a remarkable behavior during the electrocatalytic $CO_2$ reduction ($CO_2$RR), namely, the formation of metal nanoparticles from Cu single atoms, and their subsequent reversible redispersion. Here we show that the switchable nature of these species holds the key for the on-demand control over the distribution of $CO_2$RR products, a lack of which has thus far hindered the wide-spread practical adoption of $CO_2$RR. By intermitting pulses of a working cathodic potential with pulses of anodic potential, we were able to achieve a controlled fragmentation of the Cu particles and partial regeneration of single atom sites. By tuning the pulse durations, and by tracking the catalyst's evolution using *operando* quick X-ray absorption spectroscopy, the speciation of the catalyst can be steered toward single atom sites, ultrasmall metal clusters or large metal nanoparticles, each exhibiting unique $CO_2$RR functionalities.

Copper-based materials exhibit a unique ability to convert $CO_2$ into hydrocarbons and other valuable chemicals and fuels[1–3] through the electrocatalytic $CO_2$ reduction reaction ($CO_2$RR). The selectivity of copper toward a certain chemical from a broad distribution of possible reaction products, however, is hard to control. One strategy is to use Cu and nitrogen co-doped carbon materials (Cu-N-C) as catalysts, featuring singly dispersed cationic Cu species. Analogous materials based on other transition metals have been used as electrocatalysts for various processes[4–9], including the $CO_2$ conversion to CO[9–11]. Nonetheless, in Cu-N-C, the cationic Cu sites were found to be unstable under $CO_2$RR, forming metallic particles[12,13]. Surprisingly, this process is reversible, with particles redispersing upon lifting the reducing conditions. Such reversible behavior holds for a broad range of Cu-N-C catalysts, including those based on Cu phthalocyanine (CuPc)[12,14–16], or those employing different metal/covalent organic frameworks as precursors[12,13,17–19].

Here we show that the switchable nature of Cu sites provides an opportunity to form in-situ unique catalytic structures with distinct functionality. We rely on a pulsed $CO_2$RR protocol alternating between the working (cathodic) potential ($E_c$) and an anodic potential ($E_a$)[20]. By varying the durations of the cathodic and anodic pulses, and by tracking the catalyst's evolution using *operando* quick X-ray absorption fine structure (QXAFS) spectroscopy[20–22], we were able to control the average size of the Cu particles with subnanometer accuracy. Unlike previous works where the particle size was fixed by catalyst preparation[16], our approach enables steering the catalyst structure on the fly and reversibly switching between different catalytic functionalities. Thus, it allows us to explore structure-property relationships in the challenging regime of sub-nanometer particle sizes. In particular, singly dispersed cationic Cu species were shown to favor hydrogen production, ultrasmall Cu clusters yielded methane, while larger Cu

---

[1]Department of Interface Science, Fritz-Haber Institute of the Max-Planck Society, Berlin, Germany. ✉e-mail: janis@fhi-berlin.mpg.de; roldan@fhi-berlin.mpg.de

nanoparticles - CO and multicarbon products. Our findings reconcile previous reports on the high selectivity of Cu-N-C catalysts to hydrocarbons[15,23] with those on particle size effect in $CO_2RR$[24,25]. They underscore the challenge of extrapolating the structure-properties relationship derived from larger nanoparticles to ultradispersed clusters featuring just a few atoms, emphasizing that for clusters that are less than 2 nm in size, even slight variations in sizes can result in strong, non-monotonic changes in their physico-chemical properties[26–28].

## Results

### Copper particles under static and pulsed CO₂RR

Cu-N-C catalysts were prepared using an impregnation-calcination method from a ZIF-8 precursor[29,30]. Ex-situ characterization using X-ray photoelectron spectroscopy (XPS), inductively coupled plasma mass spectrometry (ICP-MS), X-ray diffraction (XRD), and high-angle annular dark-field scanning transmission electron microscopy (HAADF-STEM) confirmed the incorporation of Cu into the nitrogen-doped carbon support and lack of clusters in the as-prepared samples. See ref. 29. for the results of ex-situ characterization, and Supplementary Fig. 1 for additional HAADF-STEM images. Static $CO_2RR$ experiments were conducted in a $CO_2$-saturated 0.1 M $KHCO_3$ electrolyte at −1.35 V (bulk pH ≈ 6.8). All potential values are given with respect to the

reversible hydrogen electrode (RHE). The formation of Cu particles in Cu-N-C was monitored by QXAFS spectroscopy with a time resolution of up to 2 s per spectrum (Figs. 1, 2 and Supplementary Figs. 2 –8). X-ray absorption near edge structure (XANES) and extended X-ray absorption fine structure (EXAFS) analyses (Supplementary Note 1) agree with our previous reports[29]. Briefly, in the as-prepared Cu-N-Cs the singly dispersed $Cu^{2+}$ sites have distorted octahedral coordination (planar Cu-$N_4$ unit with two axial O or OH groups[29,30], Fig. 1a). Under applied potential, these cationic species transformed rapidly (within 100 s, Fig. 2a, d, f) into metallic particles, with an average effective diameter of ca. 1.3 ± 0.1 nm (Supplementary Note 1, Supplementary Fig. 4 and Supplementary Table 1). The metallic Cu coexists with the remaining singly dispersed Cu. The concentration of the latter after the quick initial drop changes slowly, and stabilizes at ca. 12% (Supplementary Fig. 7).

Next, in order to tune the working structure of the Cu electrocatalyst, we investigated its evolution under pulsed $CO_2RR$, with cathodic potential $E_c = -1.35$ V, and anodic potential $E_a = 0.44$ V, and different durations of the cathodic and anodic pulses ($\Delta t_c$ and $\Delta t_a$, respectively). The applied potential values were chosen so that $E_c$ is sufficiently negative for the formation of Cu clusters during the cathodic pulse, while $E_a$ is sufficiently positive for their redispersion during the anodic pulse[29]. In addition, the empirically selected pulse

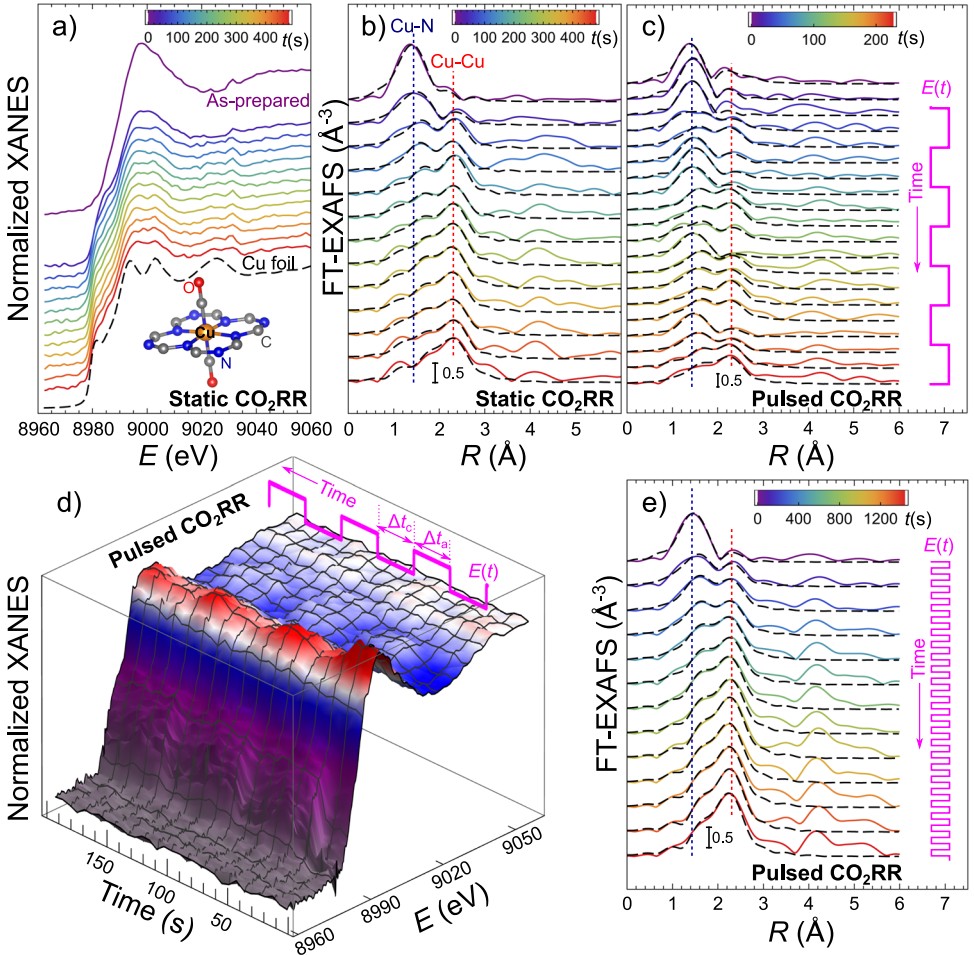

**Fig. 1 | Evolution of *operando* XANES and EXAFS spectra for Cu-N-C electrocatalysts under static and pulsed CO₂RR.** Cu K-edge XANES (**a**) and Fourier-transformed (FT) EXAFS spectra (**b**) for Cu-N-C during the first 400 s of $CO_2RR$ in 0.1 M $KHCO_3$ under static −1.35 V potential. The inset in (**a**) shows a structure model of the Cu single sites in the as-prepared catalyst[30], visualized with VESTA software[52]. Evolution of FT-EXAFS (**c, e**) and XANES (**d**) spectra during pulsed $CO_2RR$ with $E_c = -1.35$ V, $E_a = 0.44$ V and $\Delta t_a = \Delta t_c = 30$ s. **c, d** Changes in the catalyst during the

first 200 s under pulsed $CO_2RR$. **e** Changes in the catalyst during the first 1400 s under pulsed $CO_2RR$. Black dashed lines in (**b, c, e**) − EXAFS data fitting results. Vertical dotted lines in (**b, c, e**) mark the positions of the main FT-EXAFS features corresponding to the bonds between singly dispersed Cu sites and their nearest neighbors (blue line), and the Cu-Cu bonds in metallic Cu clusters (red line). Depicted spectra are averages over 20 s (**a, b**), 6 s (**c, d**) or 60 s (**e**).

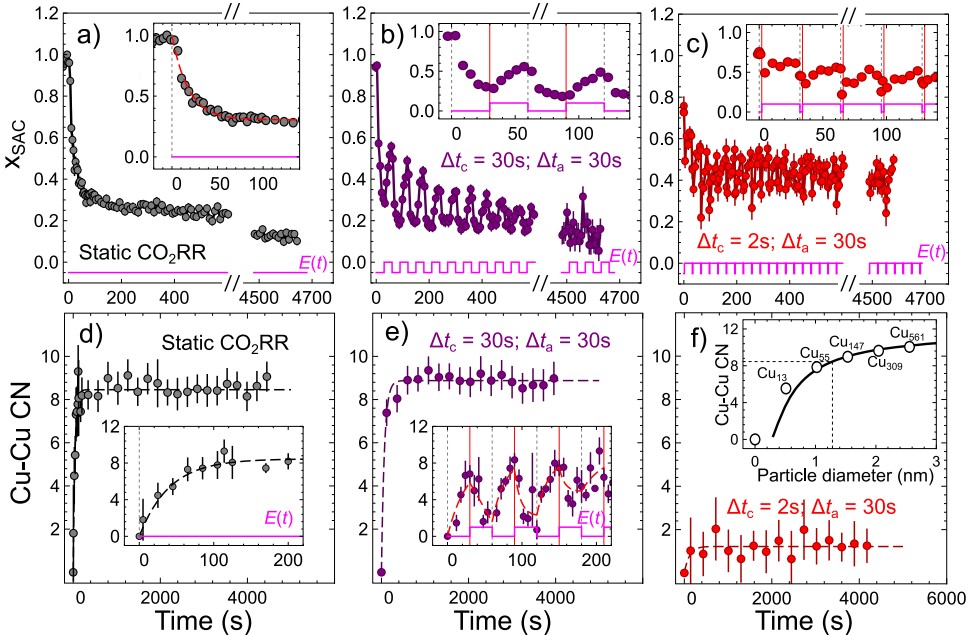

**Fig. 2 | Changes in the concentration of singly dispersed Cu sites $x_{SAC}$ and Cu-Cu coordination numbers (CNs) under static and pulsed CO₂RR.** $x_{SAC}$ values are extracted from the LCA of *operando* Cu K-edge XANES data collected under static − 1.35 V potential (**a**) and pulsed CO₂RR with $E_c = -1.35$ V, $E_a = 0.44$ V, $\Delta t_a = 30$ s and $\Delta t_c = 30$ s (**b**) or $\Delta t_c = 2$ s (**c**). Insets show zoom-ins into the first 180 s of the experiment. The red dashed line in the inset in panel (**a**) – exponential fit of $x_{SAC}$ values for the first 180 s. **d–f** corresponding changes in the Cu-Cu CNs, as extracted from the fits of *operando* Cu K-edge EXAFS. Insets in (**d, e**) show zoom-ins into the first 200 s of the corresponding experiment. Spectra used for EXAFS fitting were averaged over 228 s (**d**), 240 s (**e**), 288 s (**f**), 20 s (inset in (**d**)) or 6 s (inset in (**e**)). The depicted Cu-Cu CNs are corrected for the averaging over metallic Cu and singly dispersed Cu sites, by dividing the apparent CNs extracted from the EXAFS fit by (1-$x_{SAC}$). Black and red dashed lines in the insets in (**d, e**) are guides for the eye (exponential fits). Inset in (**f**): Relation between Cu particle size and Cu-Cu CN, constructed based on ref. 51 (solid line). Empty circles show the particle diameters and corresponding calculated Cu-Cu CNs for cuboctahedral Cu nanoparticles with 1, 13, 55, 147, 309, and 561 Cu atoms. The dotted lines in the inset in (**f**) show an example of particle size determination from the Cu-Cu CN for the final state of the Cu-N-C under static CO₂RR at − 1.35 V. Error bars show standard errors of parameter estimates.

potential values ensured comparable rates of cluster formation (under cathodic potential) and redispersion (under anodic potential). Future work is planned taking advantage of the computational design of experiment approaches to rationally select the optimum potential pulse parameters, including their shape, duration, and anodic and cathodic potential values. The largest $\Delta t_c$ value considered was 30 s, comparable to the characteristic time of Cu cluster formation under static CO₂RR (Fig. 2a–d), and, coincidentally, also with the time needed for the current equilibration upon potential change (Supplementary Fig. 9). The potential pulsing results in periodic changes in XANES and EXAFS (Fig. 1c, d), with spectral features corresponding to metallic Cu increasing during the cathodic pulse, and decreasing during the anodic pulse. We attribute the variations in XAFS under pulsed CO₂RR to the switching between the same Cu species as observed in static CO₂RR under a potential of − 1.35 V (metallic Cu species and singly dispersed cationic Cu sites). To confirm this, we relied on linear combination analysis (LCA) and principal component analysis (PCA) of *operando* XANES (Supplementary Note 2 and Supplementary Fig. 12). Specifically, both LCA and PCA show no formation of copper oxides during the anodic potential pulses.

Unsurprisingly, during the first cathodic pulse, the catalyst evolution resembles that during the first seconds under static CO₂RR. This is concluded from the LCA-XANES and EXAFS fitting in Fig. 2b–e for pulsed CO₂RR with $\Delta t_a = 30$ s and $\Delta t_c = 30$ or 2 s, and Supplementary Figs. 9, 10 for $\Delta t_c = 4$ s, 8 s and 16 s. A full list of EXAFS fitting results is given in Supplementary Tables 2–6. A subsequent anodic potential pulse increases the concentration of cationic Cu again, while gradually decreasing the particle size. For the $\Delta t_a = \Delta t_c = 30$ s, at the end of the first anodic pulse, the fraction of Cu in the cationic state reaches ca. 55%, while the Cu-Cu coordination number (CN) (corrected for the

presence of co-existing cationic Cu species) decreases to ca. 1.2, suggesting that the metallic Cu species form Cu clusters with just 2, 3 atoms. Hence, 30 s of anodic potential is not quite sufficient to fully destroy all formed Cu clusters (under the given $E_a$ value). The catalyst before the second potential cycle thus already features some ultra-small Cu clusters. The second cathodic pulse reduces again the cationic Cu and increases the size of the pre-existing Cu clusters. In the subsequent pulses, the concentration of cationic Cu and the Cu-Cu CN oscillate with a decreasing amplitude. The catalyst becomes increasingly metallic (Fig. 2b), and the average Cu-Cu CN increases (Figs. 1e, 2e) until a stationary state is reached. For shorter $\Delta t_c$ (2–4 s (Fig. 2c, f, Supplementary Figs. 10, 11)), a stationary state is reached within a few potential cycles, while larger $\Delta t_c$ (8–30 s) require ca. 5–10 min for this. In the stationary state, subtle periodic changes in the catalyst structure and chemical state persist. To better track them, we gather and average the spectra collected at the same time moments after the onset of each potential pulse (Fig. 3a).

The stationary concentration of singly dispersed Cu sites ($x_{SAC}$) oscillates around an average value that increases from ca. 18% to ca. 40% for $\Delta t_c = 30$ s and 2 s, respectively. The oscillations decrease with increasing $\Delta t_c$: The difference between minimal and maximal $x_{SAC}$ value is ca. 3% for $\Delta t_c = 30$ s, and ca. 10% for $\Delta t_c = 2$ s (Fig. 3a, b). These oscillations align with periodic changes in the Cu-Cu CNs. For $\Delta t_c = \Delta t_a = 30$ s, the Cu-Cu CN oscillates between ca. 8.0 and ca. 9.5. Similar periodic changes in CN occur for shorter $\Delta t_c$, but due to the lower signal from metallic Cu, lower amplitude of these oscillations, and shorter transformation times, these are more challenging to resolve. Nonetheless, the average Cu-Cu CN can be obtained for all $\Delta t_c$ values. It increases systematically with $\Delta t_c$ (Fig. 3a, c), reaching $1.2 \pm 0.1$ for $\Delta t_c = 2$ s, and $8.8 \pm 0.1$ for $\Delta t_c = 30$ s, showing that the size of the Cu

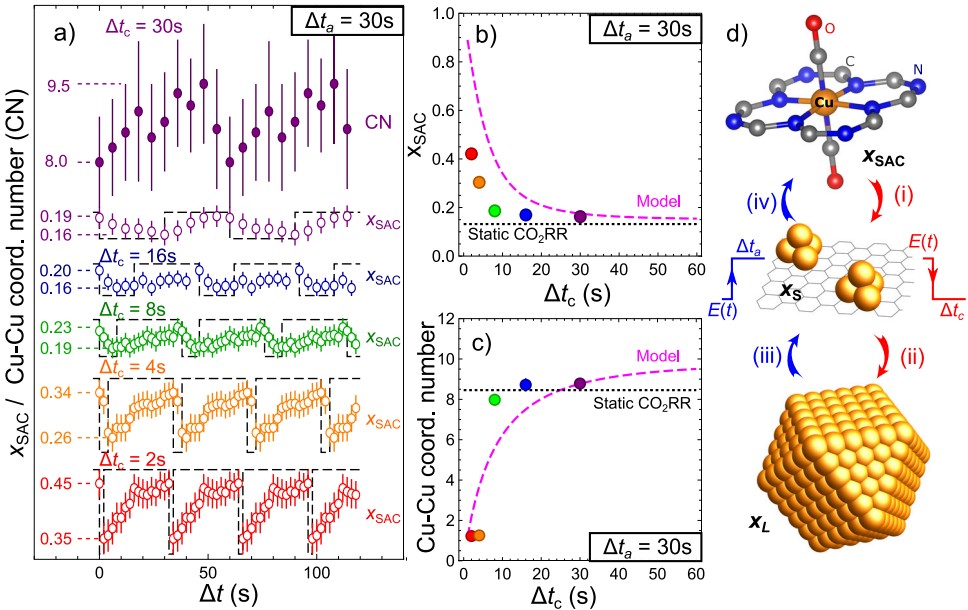

**Fig. 3 | Dependency of the concentration of singly dispersed cationic Cu species and of Cu-Cu CNs on the durations of the cathodic pulse. a** Changes in the concentration of singly dispersed Cu sites $x_{SAC}$, extracted from *operando* Cu K-edge XANES data. The spectra collected at the same time moments after the onset of each respective potential pulse, but not sooner than after 2000 s of the onset of pulsed $CO_2RR$, are averaged together. Results obtained for pulsed $CO_2RR$ with $E_c = -1.35$ V, $E_a = 0.44$ V, $\Delta t_a = 30$ s, and different $\Delta t_c$ values: 30, 16, 8, 4, 2 s are shown. For the $\Delta t_a = \Delta t_c = 30$ s case, the corresponding true Cu-Cu CNs, extracted from the same averaged data, are also shown. For clarity, all results are shifted both vertically and horizontally. **b** Average concentrations of singly dispersed cationic Cu species in the stationary state. Red, orange, green, and purple circles show the

values extracted from experimental measurements under pulsed $CO_2RR$ with $E_c = -1.35$ V, $E_a = 0.44$ V, $\Delta t_a = 30$ s, and $\Delta t_c$ values 2, 4, 8, 16, and 30 s, respectively. Results obtained under static $CO_2RR$ are also shown for comparison. The Magenta dashed line shows the dependency of the average concentrations of singly dispersed cationic Cu species on $\Delta t_c$, as derived from our model, introduced in Supplementary Note 4. **c** Corresponding true average Cu-Cu CNs. **d** Schematic representation of the main processes taking place under pulsed $CO_2RR$: conversion of singly dispersed sites into metallic species (i), particle growth (ii), fragmentation of particles (iii), and conversion of small clusters into singly dispersed sites (iv). Error bars show standard errors of parameter estimates.

---

clusters is indeed tuned by varying the durations of the potential pulses. However, caution is needed when assigning specific particle diameter values to these CNs, due to the coexistence of particles of different sizes.

## Size dependency of Cu particles on pulse parameters

The speciation of Cu-N-C under pulsed $CO_2RR$ emerges from a balance between four processes (Fig. 3d): (i) reduction of singly dispersed cationic Cu sites during the cathodic pulse (evidenced by the periodic decrease of $x_{SAC}$ in LCA-XANES); (ii) increase in the size of metallic Cu particles during the cathodic pulse (indicated by increasing Cu-Cu CNs in EXAFS); (iii) fragmentation of Cu particles into smaller clusters during the anodic pulse (evident from the decrease in Cu-Cu CN) and (iv) conversion of small Cu clusters back into singly dispersed cationic sites during the anodic pulse (shown by the periodic increase of $x_{SAC}$ in LCA-XANES).

Steps (i) and (ii) resemble the processes under static $CO_2RR$. The conversion of singly dispersed cationic sites into metallic species under cathodic potential proceeds similarly regardless of sample history (Supplementary Note 3 and Supplementary Figs. 12, 13): For fixed $\Delta t_c$, the amount of Cu single sites converted into metallic species depends only on the number of these single sites available at the onset of the cathodic pulse. Furthermore, under static $CO_2RR$ and cathodic pulse in pulsed $CO_2RR$, the Cu-Cu CN stabilizes at a value lower than that of the bulk fcc structure (12). This could be an indication that the Cu cluster growth via the migration and aggregation of the small metal particles formed in step (i) is dominating over Ostwald ripening (although both particle growth mechanisms could also coexist[31,32]). As the particle size increases, their mobility is hindered, while the inter-particle distances increase, preventing particles from further growth via diffusion-coalescence. The regeneration of atomic sites and

ultrasmall clusters (steps (iii) and (iv)), produces new mobile species, facilitating particle growth. Therefore, we observe larger Cu-Cu CN under pulsed $CO_2RR$ (ca. 9.5 for $\Delta t_c = \Delta t_a = 30$ s) than under static $CO_2RR$ (ca. 8.5).

In pulsed $CO_2RR$, the distribution of particle sizes cannot be considered to be monomodal, even in the stationary state. New ultrasmall Cu clusters are formed during each cycle, while the existing Cu clusters can grow until they reach their final size. To explain the evolution of this system, a minimal model must include particles of at least two sizes: ultrasmall Cu clusters formed in step (i), and larger Cu particles formed in step (ii). We assume that the small particles resulting from the fragmentation of larger particles (step (iii)) are similar to the clusters formed during step (i) from singly dispersed Cu. The rates of steps (i)–(iv) and the corresponding sizes of ultrasmall Cu clusters and larger Cu particles can be estimated from the changes in $x_{SAC}$ and average Cu-Cu CN during the first few potential cycles and in the stationary state for pulses with $\Delta t_c = \Delta t_a = 30$ s, Supplementary Note 4 and Supplementary Fig. 14. The model provides a semi-quantitative agreement with our experimental results (Fig. 3b, c), capturing the decrease of average $x_{SAC}$ and the increase in the average Cu-Cu CNs with increasing $\Delta t_c$. Discrepancies between the experimental results and model stem mainly from our simplified (exponential) dependencies of the transition rates on pulse durations.

Our model yields that the larger Cu particles formed under pulsed $CO_2RR$ have Cu-Cu CN ca. 10.1, corresponding to a particle diameter of ca. 2.5 nm, i.e., nearly two times larger than under static $CO_2RR$. The Cu-Cu CNs for ultrasmall Cu clusters, in turn, grow with $\Delta t_c$ and increase from ca. 0.4 for $\Delta t_c = 2$ s (suggesting the presence of single charge neutral atoms with a few dimers), to ca. 4.4 for $\Delta t_c = 30$ s, corresponding to ca. 0.6 nm particles. This is slightly lower

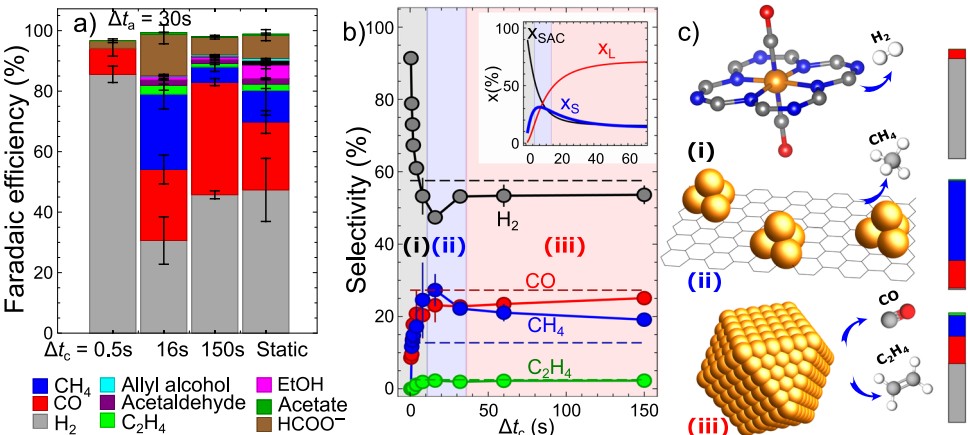

**Fig. 4 | Selectivity of Cu-N-C electrocatalyst under pulses CO₂RR with different pulse durations. a** Faradaic efficiencies of the reaction products under static CO₂RR at −1.35 V and under pulsed CO₂RR with $E_c = -1.35$ V, $E_a = 0.44$ V, $\Delta t_a = 30$ s and different $\Delta t_c$ values (30, 10, and 1 s). Uncertainties are estimated by comparing the results of at least three repeated measurements using fresh samples. **b** Corresponding selectivities for H₂ and the main gaseous CO₂RR products as a function of $\Delta t_c$. The reported values are normalized, assuming a total selectivity of 100% for all combined detected gaseous products. All measurements are performed with the same sample. Horizontal dashed lines show the corresponding selectivities observed under static CO₂RR at −1.35 V. Inset: calculated concentrations of Cu within cationic single site species ($x_{SAC}$), small Cu clusters ($x_S$), and large Cu nanoparticles ($x_L$). Shaded regions indicated as (i), (ii), and (iii) mark three distinct regimes with different selectivity trends, and dominated by different Cu species, as indicated in (**c**). Color bars in (**c**) show partial selectivities toward the main gaseous products of the respective Cu species, estimated by best-fitting the average selectivities and using the concentration profiles shown in (**b**). The same color coding is used here as in (**a**).

than the CN of 5.5 for a cuboctahedral Cu₁₃ cluster, Fig. 2f. We can also now quantify the concentrations of Cu within the different species coexisting in our sample (cationic singly dispersed species, small Cu clusters, large Cu nanoparticles, Fig. 4b, inset). We identify three distinct regions in the parameter space, dominated by different catalyst structures. For short $\Delta t_c$, Cu-N-C is dominated by the singly dispersed cationic species. The latter is the case when $\Delta t_c$ is shorter than the characteristic time for the conversion of cationic species into metallic species $\tau_{SAC\rightarrow S} = 16 s$, Supplementary Note 4. For large $\Delta t_c$, Cu-N-C is dominated by the large 2.5 nm particles. Here $\Delta t_c$ must be larger than the time for the conversion of small clusters into larger particles of $\tau_{S\rightarrow L} = 18 s$. For intermediate $\Delta t_c$, very small Cu clusters, featuring just a few Cu atoms contribute strongly. These three distinct regions also exhibit substantially different catalytic properties (*vide infra*).

In contrast to the pronounced effect of $\Delta t_c$, the changes in $\Delta t_a$ have a lesser effect on the system, Supplementary Note 5 and Supplementary Figs. 15–19. Increasing $\Delta t_a$ primarily shifts the boundaries between the three aforementioned regimes toward higher $\Delta t_c$ values.

## Catalytic selectivity

We analyzed CO₂RR products under static and pulsed conditions with $\Delta t_a = 30$ s and different $\Delta t_c$ values (Fig. 4a, Supplementary Fig. 20 and Supplementary Table 7). Under the static CO₂RR at −1.35 V, after 4000 s, the main reaction products are H₂ (with Faradaic efficiency (FE) ca. 47%), CO (FE ca. 22%), methane (FE ca. 8%) and formate (FE ca. 8%). The activity of the Cu-N-C catalyst was several times higher than that of the "bare" N-C catalyst, which features N and Zn species remaining from the synthesis[29]. The distribution of reaction products for Cu-N-C catalyst changes drastically under pulsed conditions and depends strongly on the pulse parameters. With $E_c = -1.35$ V and $E_a = 0.44$ V, for $\Delta t_c = 1$ s and lower, HER dominates over CO₂RR, reaching 85% FE, with only CO and small amounts of formate as side products. For $\Delta t_c = 10$–16 s, HER is suppressed, and we observe large amounts of CH₄ (with FE up to 25%), as well as detectable amounts of ethylene, ethanol, and acetaldehyde. Furthermore, the total current density increases ca. 5 times. Further increase in pulse duration results in a decrease in CH₄ selectivity, and an increase in H₂ and CO production.

To investigate the intriguing non-linear dependency of the selectivity toward the main gaseous products (H₂, CH₄, CO, and C₂H₄) on pulse parameters, we performed measurements for an extended range of $\Delta t_c$ values (Fig. 4b and Supplementary Figs. 21, 22). We do not consider here changes in liquid products: Fig. 4a suggests that these are rather small, and, furthermore, the quantification of these minor products under pulsed CO₂RR is challenging due to a strong contribution of non-faradaic currents (see "Methods"). To reduce the systematic errors due to differences among distinct samples, all the results in Fig. 4b were collected for the same sample, starting from the shortest pulses and exposing it to OCP for 30 min in-between different pulse conditions. An analogous approach was used to collect the aforementioned XAFS data.

The non-monotonic dependencies of the catalyst selectivity on $\Delta t_c$, as shown in Fig. 4b, are striking. Under pulses with short $\Delta t_c$ (less than 5 s, region (i) in Fig. 4b), HER dominates over CO₂RR, with hydrogen selectivity approaching 100% for the shortest $\Delta t_c$ values. Upon an increase in $\Delta t_c$, H₂ selectivity drops steeply, while the selectivities for CO and CH₄ sharply increase. A minimum of H₂ and a maximum of CH₄ selectivity is observed at ca. $\Delta t_c = 16$ s (region (ii) in Fig. 4b), where the selectivities for these two products are ca. 47% and 27%, respectively. In this region, we also observe a gradual increase in the C₂H₄, which starts to be formed at $\Delta t_c$ of ca. 2 s and plateaus at ca. $\Delta t_c = 16$ s at the selectivity value of ca 2%. A further increase of $\Delta t_c$ (region (iii) in Fig. 4b) results in a decrease of CH₄ and an increase in CO and H₂, where the latter plateaus at 53% FE. Overall, under pulsed CO₂RR for larger $\Delta t_c$, the selectivities for H₂, CO, CH₄, and C₂H₄ are stable and remarkably similar to those observed under static CO₂RR. This is, in principle, an intuitively expected behavior: the contribution of the processes occurring during the anodic pulse plays a proportionally lesser role in this regime. On the other hand, these results show that the irreversible changes in the catalyst morphology under pulsed conditions (e.g., degradation of the C-N support) have a similar effect on the catalyst's selectivity under pulsed and static CO₂RR. The importance of these irreversible changes and of the sample history is discussed in Supplementary Note 6 and Supplementary Fig. 24.

The parallels between the non-monotonic trends in the catalyst selectivity and the afore discussed changes in the catalyst structure as a function of $\Delta t_c$ allow us to plausibly rationalize the obtained results

(Fig. 4c). Specifically, in the regime (i) at low $\Delta t_c$ values, where almost all Cu is present in the form of singly dispersed cationic species, $H_2$ is the dominating product and overall faradaic currents (total currents including both HER and $CO_2$RR contributions) are low. The suppression of HER and an increase in $CH_4$ at higher $\Delta t_c$ values (regime ii), in turn, correlates well with the observed decrease in the concentration of cationic Cu and the formation of ultrasmall metallic Cu clusters. This suggests that the singly dispersed Cu sites themselves are a poor catalyst for $CO_2$RR since mainly $H_2$ was reported in (i), while the ultrasmall clusters are highly selective for methane production. Finally, at larger $\Delta t_c$ values (region (iii)), the small metallic Cu clusters are converted into larger particles, and we observe an increase in HER, a decrease in $CH_4$, and a steady production of CO and $C_2H_4$. This suggests that the large Cu particles are less selective to $CH_4$ than ultrasmall clusters while facilitating the formation of ethylene (and, likely, of other $C_{2+}$ products), Fig. 4c.

## Discussion

By decoupling the contributions of different species to the catalytic properties and sample-averaged spectroscopic data, our analysis bridges the results of previous studies and allows us to reconcile the, at times, conflicting reports on Cu particle size effects on $CO_2$RR. Specifically, the catalytic properties of our Cu-N-C in the regime where the catalyst is dominated by the contribution of larger particles, match well the $CO_2$RR selectivity trends for monodispersed particles prepared via inverse micelle encapsulation, with sizes systematically changed between 2 and 15 nm[25]. In both cases, for the 2.5 nm particles, $H_2$ is the main product, with a selectivity of 50–60%, followed by CO with a selectivity of ca 25%, $CH_4$ with a selectivity of 15–20%, and ethylene with the selectivity of a few percent. In comparison to bulk Cu catalysts, the enhanced formation of CO and hydrogen with decreasing particle size for these nanoparticles is attributed to the increased presence of step and kink sites, which facilitate the $CO_2$ and H adsorption, which are rate limiting steps for HER and for $CO_2$ conversion to CO. Moreover, as previously revealed for size-selected Au[33,34] as well as Cu nanoparticles[25], below a certain particle size, too strong hydrogen adsorption taking place on low-coordinated sites blocks CO adsorption, ultimately leading to preferred $H_2$ versus CO production. It should be, however, noted that the structure-reactivity correlations in $CO_2$RR cannot be reduced to the particle size as a single descriptor of functionality, since different structures might be stabilized for seemingly similarly-sized particles, especially given the experimental size distribution. Thus, specific details of the structure of the nanoparticles, including predominant facets and presence of defects or dislocations, which are also size-dependent, must also be considered for the understanding of the selectivity trends observed[25]. The ultrasmall Cu clusters observed in our study, in turn, exhibit drastically different catalytic behavior. Here the observed boost in $CH_4$ selectivity is remarkably consistent with the studies on CuPc-derived Cu-N-C catalysts[12,16]. In fact, in ref. 16. where the average size of Cu particles was tuned by mixing a CuPc precursor with different amounts of carbon nanoparticles, very similar non-monotonic trends in the catalyst selectivity as a function of the average particle size were observed as in our current work, despite the difference in the precursor and much higher current densities (up to 200 mA/cm²)[16]. In both cases, small Cu clusters were found to be highly selective to methane. The increase in particle size correlated with the increase in $C_2H_4$, while the increase in cationic Cu concentration enhanced HER. Based on DFT simulations, ref. 16. demonstrated that the low-coordinated Cu sites reduce the reaction energy for the hydrogenation of adsorbed CO species, explaining thus the high selectivity of ultradispersed Cu clusters to methane, which we also observe in our current work. We emphasize, however, that besides confirming the previous findings, our work provides a practical recipe for switching between the distinct catalytic regimes,

and demonstrates the possibility of robustly fine-tuning the average particle sizes with subnanometer accuracy by simply adjusting the $\Delta t_c$ value, leading to the possibility to also steer the distribution of reaction products on demand. More importantly, it provides a pathway for understanding the fundamental properties of these distinct catalytically active Cu species that we can now controllably (re-)generate through a careful selection of the dynamic reaction conditions. Thus, the main outcome of our work is the demonstration of a catalyst operation method that allows exploring structure-reactivity correlations within a large structure parameter space. One should, nonetheless, acknowledge that in addition to the dynamic changes in particle size and overall structure, other effects taking place under pulsed $CO_2$RR conditions could also have an impact on the observed catalytic trends, such as irreversible changes in the carbon support, periodic changes in the adsorbate structures, and variations in the local pH and concentrations of reactants and intermediates[20,35].

We have shown that the upper size limit for Cu particles in the Cu-N-C system, both under static and pulsed $CO_2$RR, is determined by the low mobility of the large Cu particles on the N-C support. Under pulsed $CO_2$RR the particles of larger sizes could be generated through the periodic regeneration of mobile ultradispersed species, but the maximal achievable particle size remains still limited to a few nm. The strong interactions between the particles and the N-C support that restricts the particle growth might also play a decisive role in the fragmentation of the Cu particles under anodic potential, as has been proposed in several works[12,36]. The absence of Cu oxide species in our system, even under open circuit conditions or anodic potential, suggests that the interactions with N-C support make the small oxide nanoparticles to be thermodynamically unfavorable[12]. This results in the quick destruction of the Cu particles, once the reducing conditions are lifted. Our results do not support the alternative hypothesis, namely, that the dissolution of Cu species in the electrolyte could play a role in the disappearance of the metallic Cu particles[36,37]. In all our cases and operation conditions, the loss of the catalyst due to the dissolution of Cu is less than 30% in 5 h experiments, according to the changes in Cu K-edge fluorescence signal, and under pulsed $CO_2$RR we find no correlation between the concentration of Cu, as tracked by Cu fluorescence signal intensity, and the appearance or disappearance of the Cu clusters (Supplementary Fig. 8).

In summary, our study demonstrated that a high degree of control of the structure and composition of electrocatalysts can be achieved via pulsed electrolysis. In particular, *operando* QXAFS served to monitor the drastic and reversible structural transformations of Cu-N-C during $CO_2$RR and to disentangle the contribution of ultradispersed metal species, clusters, and nanoparticles. Moreover, we revealed how to use pulsed electrolysis to generate such species on demand, which allowed us to rationalize the convoluted structure-property relationship during $CO_2$RR. Here, the insight from time-resolved *operando* spectroscopy played a key role in identifying the regions within a vast pulse parameter space, where distinct catalyst structures and compositions and, hence, catalytic functionalities are expected. We were thus able to tune the $CO_2$RR selectivity from mainly $H_2$ production to $CH_4$ or higher-order hydrocarbons. We envision that such an approach can be beneficial for other electrocatalytic processes as well, e.g., for the electrochemical nitrate reduction, where similar transformations of Cu-N-C catalysts are believed to also take place during operation[19]. Furthermore, similarities in the dynamic processes occurring in Cu-N-C with the reversible exsolution of metal particles from oxide supports[38–40], or with the reversible formation/fragmentation of metal nanoparticles encapsulated in zeolite structures[41,42] hint that the oxidative/reductive pulse approach presented here might also serve to control the size of the catalyst particles formed in thermal catalysis applications.

## Methods

### Sample preparation

ZIF-8 framework was synthesized using 2-methylimidazole (4.92 g, 99% Sigma-Aldrich) and zinc-nitrate hexahydrate (4.24 g, 98%, Acros Organics), dissolved in methanol (500 mL). This solution was heated to 60 °C and stirred for 24 h under reflux, followed by centrifugation and thorough washing with methanol and ethanol. The resulting powder was dried at 60 °C in air. The obtained ZIF-8 powder was pyrolyzed for 1 h in Ar flow (100 mL min$^{-1}$) at 1000 °C, resulting in the metallization and evaporation of Zn species within the ZIF-8 framework. The remaining porous N-doped carbon structure was acid-washed for 24 h in 20 w% HNO$_3$ ($\geq$ 65%, Carl Roth). The acid treatment removes the remaining crystalline Zn-containing species. The resulting N-C precursor sample was washed thoroughly with ultrapure water.

To prepare Cu-N-C catalysts, the N-C precursor was impregnated with Cu$^{2+}$ species. For this purpose, N-C was dispersed in 6 mM Cu(NO$_3$)$_2$·3 H$_2$O (Sigma-Aldrich, >99%) solution in isopropanol ($\geq$ 99.8%, Sigma-Aldrich) by sonicating for 2 h at ~40 °C. The suspension was then stirred for another 2 h at room temperature (RT). The Cu-N-C precursor was collected by centrifugation, air dried at 60 °C and heated in Ar (100 mL min$^{-1}$) at 700 °C for 1 h. The obtained Cu-N-C was stirred in 20 w% HNO$_3$ for 24 h and washed with ultrapure water. Finally, the catalyst was dried in air at 60 °C.

### Electron microscopy

STEM images were obtained using a probe-corrected JEM-ARM 200 F (JEOL, Japan) operated at 200 kV and equipped with a cold field emission gun (CFEG). HAADF signals were acquired from an electron probe with a 14.2 mrad convergence semi-angle and a 90–370 mrad collection semi-angle. The beam current was kept at 11 pA, resulting in an electron dose scaled by the pixel size. For the samples after the reaction, the catalysts were collected after 4000 s under static or pulsed CO$_2$RR. The samples were removed from the glassy carbon by sonication in ethanol and drop-casted onto Au TEM grids.

### Operando QXAFS measurements

The main XAFS spectra measurements in QXAFS mode were carried out at the SuperXAS beamline at the SLS synchrotron (Villigen, Switzerland) using a LN-cooled channel-cut Si(111) monochromator for energy selection[43]. The monochromator was oscillating with 1 Hz frequency. The intensity of the incoming X-rays was measured using an N$_2$-filled ionization chamber. Rh-coated collimating and focusing mirrors were used to reduce the heat load and reject higher harmonics. XAFS measurements were performed in fluorescence mode at the Cu K-edge (8979 eV) using a PIPS detector.

*Operando* QXAFS data for pulsed CO$_2$RR with $\Delta t_a = \Delta t_c = 30$ s were collected at the P64 beamline at PETRA III (Hamburg, Germany)[44,45] using settings similar to those at SLS SuperXAS.

Supplementary *operando* XAFS data for pulsed CO$_2$RR ($\Delta t_a = 1$ s, $\Delta t_c = 1$ s or 4 s; and $\Delta t_a = 4$ s, $\Delta t_c = 1$ s or 4 s) were collected at CryoEX-AFS endstation at KMC-3 beamline at BESSY II synchrotron (Berlin, Germany)[46]. In these measurements the acquisition of each XAFS spectrum took several minutes, therefore, the analysis of these data shows the chemical state and particle sizes, averaged over many potential pulses. The Cu K-edge fluorescence data were collected using a 13-element Si drift detector.

For *operando* measurements, we used an in-house built single-compartment cell (Supplementary Fig. 25)[47]. The samples were spray-coated on a carbon electrode, which was used as a working electrode while acting also as a window for the incident and fluorescent X-ray photons. A Pt mesh was used as a counter-electrode, and a leak-free Ag/AgCl electrode was used as a potential reference. Measurements were done in a CO$_2$-saturated 0.1 M KHCO$_3$ electrolyte (pH = 6.8), with CO$_2$ continuously bubbling through the cell. The electrolyte volume in the cell is 35 ml. Circulation of the electrolyte was ensured by a peristaltic pump. The potential was controlled by a *BioLogic* potentiostat. The potentials reported in the text are given with respect to the reversible hydrogen electrode (RHE), using the following equation:

$$E_{RHE} = E_{Ag/AgCl} + 0.242\,V + 0.059 \cdot pH. \tag{1}$$

For the calibration of the QXAFS data, at the beginning of each scan a Cu foil spectrum was collected in transmission mode. Calibration of the QXAFS data was done using beamline-specific software[48]. Spectra averaging, background subtraction, and XANES data processing were carried out using a set of in-house built Wolfram Mathematica scripts. Extraction and fitting of EXAFS data was carried out using LARCH[49] and FEFFIT[50] codes.

For averaging XAFS spectra collected at the same time moments after the onset of each respective potential pulse (Fig. 3a), XAFS spectra μ($t$) with the same φ = ($t$ mod $T$) are gathered and averaged, where $t$ is the time at which the respective spectrum was collected (with $t = 0$ corresponding to the onset of pulsed CO$_2$RR), $T = \Delta t_c + \Delta t_a$ is the total duration of potential cycle, and "mod" denotes modulo operation. Only the spectra corresponding to the steady state, i.e., collected a sufficiently long time after the onset of the pulsed CO$_2$RR, are included in the averaging ($t > 2000$ s).

### Interpretation of XANES and EXAFS data

Linear combination fits of XANES spectra were performed using the Cu K-edge XANES spectrum for a Cu foil and a spectrum for the as-prepared Cu-N-C catalyst as references. This approach thus neglects the dependency of the XANES spectra for metallic Cu on the particle size, and the changes in the XANES spectra for singly dispersed Cu species during the exposure of these sites to CO$_2$RR conditions. The concentration of singly dispersed Cu sites $x_{SAC}$ can be estimated from LCA-XANES results simply as the weight of the corresponding reference spectrum in the linear combination. The concentration of metallic Cu is then given by $1 - x_{SAC}$.

Fitting of the EXAFS spectra χ($k$)$k^2$ was carried out in $R$-space in the range from 1.0 up to 2.8 Å (up to 2.0 Å for samples that did not contain metallic Cu clusters). Fourier transform was carried out in the $k$ range from 2.0 up to 8.0 Å$^{-1}$. We included in the fit Cu-O and Cu-Cu paths. Here, Cu-O paths account also for the possible Cu-C and Cu-N bonds, since these cannot be discriminated easily by EXAFS analysis. For both paths, their average interatomic distance $R$, coordination number $N$, and disorder factor σ$^2$ were refined. Furthermore, the correction to the photoelectron reference energy $\Delta E_0$ was also fitted. Amplitude reduction factors $S_0^2 = 0.85$ for the Cu-Cu bond and 0.68 for the Cu-O bond were determined from the EXAFS fits of reference materials (Cu foil and CuO).

The apparent coordination numbers obtained in EXAFS fits $N_{Cu-O}$ and $N_{Cu-Cu}$ are averages over all Cu species in the sample. Since singly dispersed Cu sites do not have another Cu atom in their first coordination shell, and the metallic Cu species are not expected to form strong bonds with C, N, or O atoms, this allows us to obtain an independent estimate of the concentrations of singly dispersed and metallic Cu. Assuming that singly dispersed Cu sites remain 6-coordinated during the CO$_2$RR, the concentration of singly dispersed Cu sites can be estimated as $x_{SAC} = N_{Cu-O}/6$. The validity of this assumption can be questioned. However, the observed good agreement between $x_{SAC}$ values obtained independently from XANES and EXAFS data analyses validates our approach.

Furthermore, the true Cu-Cu coordination numbers for the metallic Cu phase can now be obtained as $\widetilde{CN}_{Cu-Cu} = CN_{Cu-Cu}/(1 - x_{SAC})$. The latter quantity is directly linked to the particle's volume-to-surface ratio, and, hence, the average particle size[47]. Following one simple approach, the relation between the size of the spherical particle with fcc-type structure and true metal-metal coordination number can be approximated with the following

formula[51]:

$$\frac{\widetilde{CN}_{Cu-Cu}}{CN_{fcc}} = \left[1 - \frac{3}{4}\left(\frac{R}{R_{nano}}\right) + \frac{1}{16}\left(\frac{R}{R_{nano}}\right)^3\right] \qquad (2)$$

Here, $R$ is the interatomic distance (2.56 Å for copper), $R_{nano}$ is the radius of the particle, and $CN_{fcc} = 12$ is the coordination number in the corresponding bulk metal (see Fig. 2f, inset).

### Measurements of electrocatalytic properties

To extract the electrocatalytic properties, an H-type cell with two compartments separated by an anion exchange membrane (Selemion, AMV, AGC Inc.) was used. The membrane is stored in a wet state, and prior to the measurement, it was rinsed with water. One compartment contains the working electrode, where both sides of a $0.5\,cm^2$ carbon paper piece (Toray Carbon Paper, GGP-H-60) were exposed to the electrolyte, while only one side was spray-coated with the catalyst, making a geometric area of $0.5\,cm^2$. A leak-free Ag/AgCl reference electrode (LF-1, Alvatek) is placed near the working electrode. The reference electrode was checked against a reversible hydrogen electrode (Gaskatel) in $CO_2$ sat. 0.1 M $KHCO_3$, pH 6.8, see Eq. (1), and a possible drift in the potential of the Ag/AgCl reference electrode was accounted for. The counter electrode in the second compartment consists of a platinum gauze (MaTecK, 3600 mesh $cm^{-2}$). A defined amount of previously purified (cation-exchange resin, Chelex 100 Resin, Bio-Rad) 0.1 M $KHCO_3$ (Alfa Aesar, 99.7%) was filled into the cell and saturated with $CO_2$ (4.5 N) with a constant flow of 20 ml $min^{-1}$ during $CO_2RR$. $CO_2RR$ was performed with an Autolab potentiostat (PGSTAT 302 N, Metrohm). The acquired data was collected with the Nova 2.1.5 software and processed via OriginPro®2023b software. The electrolyte was prepared immediately before the experiment and saturated with $CO_2$ for at least 15 min. Ohmic drop resistance, usually between 6 and 14 Ω, was measured via the $i$-interrupt method prior to the reaction. The $CO_2$ flow towards both the cathode and the anode compartment was controlled with a mass flow controller (Bronkhorst, EL-Flow). The correct flow was ensured with a flow meter (Agilent, ADM, G66991A) to be 20 ml/min prior to each measurement.

Online gas product detection was performed after 60 s and every 15 min during $CO_2RR$ with a gas chromatograph (GC, Agilent 7890B) for a total time of 4000 s at each pulse condition. Our GC featured a thermal conductivity detector (TCD) for $H_2$ detection and a flame ionization detector (FID) for carbon products. The liquid products were detected with a liquid GC (L-GC, Shimadzu 2010plus), equipped with a fused silica capillary column and an FID detector. Acetate and formate were detected with high-performance liquid chromatography (HPLC, Shimadzu Prominence), equipped with a NUCLEOGEL SUGAR 810 column and a refractive index detector (RID).

Measurements were done by using a fresh sample for each pulse condition (when both liquid and gaseous products were quantified, Fig. 4a), or by reusing one sample several times with one pulse sequence following another, with 30 m rest at open circuit potential in between. The electrolyte was not changed in between the different pulse conditions, thus, only gaseous products were monitored. Here, the selectivity and partial current densities are reported instead of the Faradaic Efficiency.

The measured currents are dominated by non-Faradaic contributions for pulses with $\Delta t_c$ values lower than 10 s, hindering the systematic evaluation of the charge provided for the Faradaic processes. Nonetheless, the currents during the cathodic pulses for all $\Delta t_c$ values match well with each other over the whole course of pulsed $CO_2RR$ (Supplementary Fig. 21), allowing the assumption that both, the capacitive and the faradaic contributions are comparable. Even though the capacitive current dominates the currents of very short pulse durations, non-neglectable amounts of products were found.

The Faradaic efficiency of each product was calculated by adapting the previously reported procedure[20]. In order to deconvolute the capacitive from the faradaic currents, specifically for very short cathodic pulses where capacitive currents dominate over the faradaic currents, we first assumed that the detected products sum up to a total Faradaic efficiency of 100% ($FE_{total,ass.}$). Thus, the partial current density $j_{partial,x}$ for each product $x$ was calculated as:

$$j_{partial,x} = \frac{\dot{V}\,C_x\,z_x\,F}{A\,V_M\,\frac{\Delta t_c}{\Delta t_c + \Delta t_a}\,FE_{total,ass.}} \qquad (3)$$

with

$\dot{V}$: $CO_2$ gas flow rate / L $s^{-1}$
$C_x$: Volume-fraction of the product $x$ detected by GC
$z_x$: Electrons transferred for reduction to product $x$
$F$: Faradaic constant / C $mol^{-1}$
$A$: Geometric area of the electrode / $cm^{-2}$
$V_M$: Molar volume / 24.5 l $mol^{-1}$
$\frac{\Delta t_c}{\Delta t_c + \Delta t_a}$: Factor to account for the absolute time actual $CO_2RR$ while pulsing
$FE_{total,ass.}$: assumed total Faradaic Efficiency / %.

Afterwards, the obtained total current densities were used to calculate the Faradaic Efficiency of each product $x$ as:

$$FE_x = \frac{j_{partial,x}}{\sum_{i=x}^{j} j_{partial,x}} *100 \qquad (4)$$

$FE_x$: Faradaic Efficiency of product $x$
$j_{total}$: Total current density, calculated as $\sum_{i=x}^{j} j_{partial,x}$ / A $cm^{-2}$

FEs for liquid products were calculated as discussed in our previous work[20].

## Data availability

Catalytic activity data processed XAS spectra, and EXAFS fitting results are provided in the Supplementary Information. The numerical values for the data shown in the Figures in the main text are provided in the attached Source Data file. The raw XAS data (requiring specialized software to be processed), and all other data that support the findings of this study are available from the corresponding authors on request. Source data are provided with this paper.

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

## Acknowledgements

We acknowledge the Paul Scherrer Institute, Villigen, Switzerland, and DESY (Hamburg, Germany), a member of the Helmholtz Association HGF, for the provision of experimental facilities. We would like to thank Dr. A. Clark and Dr. M. Nachtegaal for their assistance in using the SuperXAS beamline of the SLS and Dr. W. Caliebe for assistance in using the P64 beamline of PETRA III. Supporting XAS measurements were also carried out at the KMC-3 XPP instrument at the BESSY II electron storage ring operated by the Helmholtz-Zentrum Berlin für Materialien und Energie, and we thank Dr. G. Schuck and Dr. M. Haumann for the support during these measurements. DH thanks the funding provided by the Alexander von Humboldt Foundation.

## Author contributions

J.T. and B.R.C. designed the study and co-wrote the paper. B.R.C. supervised the study. J.T. designed and supervised the *operando* XAS experiments and analyzed XAS data. C.R. designed and performed catalytic activity studies. D.H. performed sample synthesis and characterization, and performed initial catalytic activity studies. E.O. performed STEM measurements. J.T., D.H., M.R., A.H., T.W., A.B., U.H., A.Y., A.M., E.L., and M.M. participated in the synchrotron experiments. All authors contributed to the discussion and interpretation of the results and editing of the manuscript.

## Funding

## Competing interests

The authors declare no competing interests.
