## [Peer Review file · Nature Communications]

REVIEWER COMMENTS

Reviewer #1 (Remarks to the Author):

Janis et al. present a compelling investigation elucidating the intricate relationship between the structural features and composition of Cu species and their resultant product distribution in the context of CO₂ reduction reaction (CO₂RR). The authors employ detailed operando X-ray absorption spectroscopy (XAS) experiments to explicitly characterize the structure and composition of Cu species, marking a significant advancement within the CO₂RR community. The observed size-dependent product distributions offer valuable insights into the design principles governing electrocatalysts, thereby paving the way for further advancements in CO₂RR technology. While the catalytic performance and product selectivity warrant additional optimization, the establishment of a robust structure-function relationship through operando XAS studies deserves commendation for its contribution towards advancing dynamic studies in CO₂RR. Consequently, I recommend the publication of this manuscript pending the following revisions.

1. Although the study successfully demonstrates size-dependent product distributions in CO₂RR, it is imperative to delve deeper into the underlying reasons driving these variations among different sizes of Cu catalysts. Specifically, elucidating why Single Atom Catalysts (SACs) exhibit a propensity towards hydrogen production, while metal clusters tend to favor methane or even higher hydrocarbons, is crucial. Concrete evidence needs to be added to establish the connection between catalyst size and the generation of different products. Furthermore, it is essential to explore whether the size effect alone governs the observed product distributions or if changes in coordination environment also play a role. Thus, supplementary experiments are suggested to provide a more comprehensive understanding of the mechanisms influencing product selectivity.

2. While the manuscript effectively demonstrates the dynamic processes during the reduction potential, the mechanism underlying catalyst oxidation remains less explicit. For instance, the potential formation of copper oxide necessitates careful consideration. Given the limitations of XAS in distinguishing between Cu-N and Cu-O bonds, additional characterizations are advised to strengthen the conclusions. Notably, preventing the formation of copper oxide is vital for the regeneration of Cu SACs, highlighting the significance of further investigations into catalyst oxidation mechanisms.

3. The rationale behind selecting specific cathode and anode potentials to control catalyst size presents an intriguing aspect of the study. Providing insights into the optimization of these potentials would enhance the manuscript's completeness and relevance.

4. To substantiate claims regarding the size of Cu catalysts, it is recommended to include corresponding High-Angle Annular Dark Field Transmission Electron Microscopy (HAADF-TEM) images depicting different stages of catalyst growth. This addition would bolster the clarity and credibility of the findings.

In conclusion, this work has made notable strides in unraveling the intricate dynamics of CO₂RR, with their study underscoring the importance of structural characterization in tailoring catalyst design. Addressing the aforementioned points will undoubtedly strengthen the manuscript and contribute to its significance within the field.

Reviewer #2 (Remarks to the Author):

In this work, the authors applied pulsed CO₂RR protocol and operando quick XAS spectroscopy to study the behaviour of single-Cu-sites catalyst. They found that the Cu particle size can be rationally tuned by the pulsed electrolysis, which was confirmed by the operando quick XAS. Combining with the CO₂RR results, they investigated the relationship between Cu size and product selectivity. Overall, the topical is interesting, well done and the manuscript is well organized. Revision is recommended.

Comments:

1. Although I appreciate that the focus of this work is on the fundamental catalysis, I worry that the outcomes may not be applicable to practical systems. Specifically - could a catalyst system that is this dynamic, be stable for 1000 hours? The < 30% loss over 5 hours is notable, but this is still a lot of catalyst loss. I don't expect the authors to generate 1000h stability, but the MS would be strengthened if this question could be addressed directly - perhaps in the discussion.

2. The author assumed four processes during the pulse CO₂RR, (i) reduction of singly dispersed cationic Cu sites during the cathodic pulse; (ii) increase in the size of metallic Cu particles during the cathodic pulse; (iii) fragmentation of Cu particles into smaller clusters during the anodic pulse, and (iv) conversion of small Cu clusters back into singly dispersed cationic sites during the anodic pulse. Does the Cu species formed in each step, either particle or cluster or sites, possess the same local coordination structure? This is of particularly importance, as the coordination environment influences activity. To solve this question, more analysis on the XAFS data and more characterization is required.

3. More characterization is needed for the Cu samples post pulse electrolysis, such as HRTEM, XRD, which will give direct information on the particle size.

4. It is hard to believe that, when the single-Cu-site regenerated under anodic potential, namely “mobile ultradispersed species”, it can remain mobility on the electrode surface, yet not be dissolved into the electrolyte, especially considering that Cu is readily corroded under an anodic potential. This needs quantitative and direct support.

5. The authors obtained slightly larger Cu-Cu CN of “ $\Delta t_c = \Delta t_a = 30s$ ” sample than that of “static” sample. However, the CO₂RR shows a big difference between “ $\Delta t_c = 150s$ ” and “static” sample, especially if the C₂/C₁ ratio is calculated. Explain and support.

5. Detailed fitting parameters should be given for all the EXAFS data fitting, including the R-factor, Debye-Waller factor, and so on. Furthermore, if we simply compare the Figure S2 and S3, it seems the k-space data used in the data fitting has better signal-noise ratio. Did the author redo the spline? More details should be provided in the method part.

6. Comment - I appreciated the synthesis re. single-atom Cu (not good at CO₂R!), vs. small clusters, vs. larger. This is useful and I agree.

Overall, a paper of interest, worthy of revision.

Reviewer #3 (Remarks to the Author):

Reviewer #4 (Remarks to the Author):

Single site metal catalysts have been a hot topic in electrocatalysis due to their unique properties of catalytic activity and selectivity. In CO₂ electroreduction, many interesting results have been obtained via single site catalysts of Ni, Co, Fe and especially Cu.

In the literature, it has been discovered that Cu was not stayed as single sites during CO₂ electroreduction, it was reduced and form nanocluster or particle during electrolysis. However there are still many unclear points about this process. Cuenya and her group have further studied on this Cu single site catalyst and their nanoparticle formation. By varying the durations of the cathodic and anodic pulses, and by tracking the catalyst's evolution using operando quick X-ray absorption fine structure spectroscopy, they were able to control the average size of the Cu particles with subnanometer accuracy. In particular, they also found that, singly dispersed cationic Cu species were shown to favor hydrogen production, ultrasmall Cu clusters yielded methane, while larger Cu nanoparticles - CO and multicarbon products.

This manuscript should be considered for a publication on Nature Comm after a revision.

1. the author should be more clear about the function of Zn in the materials since Zn is also well known as a selective catalyst for CO₂ reduction to CO.
2. The formation of nanoparticle by electrocatalysis of CO₂, this nanoparticle is only Cu or bimetallic CuZn. It need to be studied more and make it clear.
3. Via XANES analysis, it is clear that Cu-N has been transformed into Cu-Cu during catalysis. Is there any existence of Cu in the reduction process?

Point-by-point response to the Reviewers' comments

The Reviewers' comments are always followed by our responses. **The action taken is highlighted in yellow.**

Reviewer #1:

Janis et al. present a compelling investigation elucidating the intricate relationship between the structural features and composition of Cu species and their resultant product distribution in the context of CO₂ reduction reaction (CO₂RR). The authors employ detailed operando X-ray absorption spectroscopy (XAS) experiments to explicitly characterize the structure and composition of Cu species, marking a significant advancement within the CO₂RR community. The observed size-dependent product distributions offer valuable insights into the design principles governing electrocatalysts, thereby paving the way for further advancements in CO₂RR technology. While the catalytic performance and product selectivity warrant additional optimization, the establishment of a robust structure-function relationship through operando XAS studies deserves commendation for its contribution towards advancing dynamic studies in CO₂RR. Consequently, I recommend the publication of this manuscript pending the following revisions.

1. Although the study successfully demonstrates size-dependent product distributions in CO₂RR, it is imperative to delve deeper into the underlying reasons driving these variations among different sizes of Cu catalysts. Specifically, elucidating why Single Atom Catalysts (SACs) exhibit a propensity towards hydrogen production, while metal clusters tend to favor methane or even higher hydrocarbons, is crucial. Concrete evidence needs to be added to establish the connection between catalyst size and the generation of different products. Furthermore, it is essential to explore whether the size effect alone governs the observed product distributions or if changes in coordination environment also play a role. Thus, supplementary experiments are suggested to provide a more comprehensive understanding of the mechanisms influencing product selectivity.

Our response: We thank the Reviewer for his/her positive comments. To explain the different catalytic properties of the different species observed in our catalysts, we first note that it is long established that the active species for CO₂ conversion to hydrocarbons over Cu-based catalysts are metallic Cu species, due to their optimal binding energies to reaction intermediates, in particular, optimum binding of CO versus hydrogen. Therefore, it is not surprising that the cationic singly dispersed Cu sites cannot contribute to the formation of methane, and even less so – to the production of more complex C₂₊ products, which requires the presence of two neighboring Cu sites to enable C-C coupling. Furthermore, the catalytic properties that we observe for larger Cu nanoparticles (with sizes of ca 2 nm) are fully consistent with the previous reports on particle size effects on CO₂RR [Reske et al, J. Am. Chem. Soc. 136, 6978 (2014) (Ref[25] in this manuscript) and Mistry et al, J. Am. Chem. Soc. 136, 16473 (2014) (new Ref.[34] in the manuscript) and Varo et al, J. Am. Chem. Soc. 146 2015 (2024) (new Ref.[35])]. The mechanistic details behind these selectivity trends are explained in detail in Ref.[25]. In particular, the high selectivity of these particles to CO and H₂ (in comparison to bulk Cu) is explained by the presence of step and kink sites on the nanoparticle surface, which enhance the adsorption of CO₂ and H on the catalyst surface,

which is the rate determining step both for CO₂ conversion to CO and for HER (Ref.[25]). For size-controlled Au nanoparticles, a size dependent trend was also found (Ref.[34], Ref.[35]) with decreasing CO production and increasing H₂ with decreasing NP size. Here, density functional theory calculations showed that these trends are related to the increase in the number of low-coordinated sites on small NPs, which favors the evolution of H₂ over CO₂ reduction to CO. We have added this point to the manuscript. Finally, the most interesting finding of our study is related to the enhanced CH₄ formation, observed for ultrasmall Cu clusters. However, this effect was already anticipated by DFT simulations in [Xu. et al, Nature Commun. 12, 2932 (2021) (Ref.[16] in this manuscript)]. In particular, it is shown in Ref.[16] that the reaction energy for the hydrogenation of the adsorbed *CO species (a necessary step in the CO₂ conversion to methane) is significantly reduced for strongly undercoordinated Cu sites.

While the changes in particle sizes are likely to be the most decisive factor for the observed trends in the catalytic properties, we, nonetheless, completely agree with the Reviewer that under pulsed reaction conditions other effects can also contribute to the catalytic function. For example, the effect of dynamic (reversible) changes in the catalyst structure coexists with other irreversible changes, such as the possible modification of the carbon support. We discuss the role of these irreversible changes in the Supplementary Note 6. The potential pulsing results also in changes of the adsorbate structures, modulation of local pH and CO₂ concentrations in the electrolyte, all of which could affect the activity and selectivity of the catalyst. We have emphasized this point better in the revised version of the manuscript, and added a new reference to a recent review paper by Casebolt et al [Casebolt et al, Joule 5, 1987-2026 (2021) (new Ref.[36] in the manuscript)] where these effects are discussed in detail.

Changes to manuscript:

To the “Discussion” section, we have added the following sentence: *“In comparison to bulk Cu catalysts, the enhanced formation of CO and hydrogen with decreasing particle size for these nanoparticles is attributed to the increased presence of step and kink sites, which facilitate the CO₂ and H adsorption, which are rate limiting steps for HER and for CO₂ conversion to CO. Moreover, as previously revealed for size-selected Au as well as Cu nanoparticles, below a certain particle size, too strong hydrogen adsorption taking place on low-coordinated sites blocks CO adsorption, ultimately leading to preferred H₂ versus CO production.*

It should be, however, noted that the structure-reactivity correlations in CO₂RR cannot be reduced to the particle size as a single descriptor of functionality, since different structures might be stabilized for seemingly similarly-sized particles, especially given the experimental size distribution. Thus, specific details of the structure of the nanoparticles, including predominant facets, presence of defects or dislocations, which are also size-dependent, must also be considered for the understanding of the selectivity trends observed. ”

We have also added:

“One should, nonetheless, acknowledge that in addition to the dynamic changes in particle size and overall structure, other effects taking place under pulsed CO₂RR conditions could also have an impact on the observed catalytic trends, such as irreversible changes in the

carbon support, periodic changes in the adsorbate structures and coverage, and variations in the local pH and concentration of reactants and intermediates.”,

and new references (Casebolt et al, Joule 5, 1987-2026 (2021) (new Ref.[36]), Mistry et al, J. Am. Chem. Soc. 136, 16473 (2014) (new Ref.[34]) and Varo et al, J. Am. Chem. Soc. 146 2015 (2024) (new Ref.[35]).

2. While the manuscript effectively demonstrates the dynamic processes during the reduction potential, the mechanism underlying catalyst oxidation remains less explicit. For instance, the potential formation of copper oxide necessitates careful consideration. Given the limitations of XAS in distinguishing between Cu-N and Cu-O bonds, additional characterizations are advised to strengthen the conclusions. Notably, preventing the formation of copper oxide is vital for the regeneration of Cu SACs, highlighting the significance of further investigations into catalyst oxidation mechanisms.

Our response: While we agree with the Reviewer that distinguishing the Cu-N and Cu-O bonds by conventional EXAFS fitting is challenging, we note that the oxides and the singly dispersed Cu-N₄ structures can be easily distinguished by XANES analysis, since the local electronic and atomistic structures are expected to be different in these cases. The principal component analysis of XANES spectra (see Supplementary Note 2 and Figure S12), suggests strongly that under pulsed reaction conditions only two spectroscopically distinct species are present in our catalyst, namely, metallic Cu and cationic copper (Cu²⁺) species with the same local structure as the as-prepared catalyst. This analysis allows us to rule out the formation of oxide structures in our catalyst. Furthermore, we note that under the anodic potential of +0.44V_{RHE} used in this work, for bulk Cu catalyst we would expect oxidation to Cu⁺ state, rather than to Cu²⁺ state [Timoshenko, J. et al. Nature Catalysis 5, 259 (2022). (Ref.[20] in this manuscript)]. Cu⁺ species can be easily distinguished from Cu²⁺ by XANES analysis, but also from EXAFS fitting, where the presence of Cu(I) oxide would result in much shorter Cu-O longer bonds than observed in our work. To address this important point raised by the reviewer, we have now emphasized it better in the revised manuscript. Thus, based on our XANES and EXAFS data, we can confidently rule out the presence of oxide species in our catalyst.

Changes to manuscript:

In Supplementary Note 2, we have added *“We also note that the formation of Cu(I) oxide species (the most likely oxide species under $E_a = 0.44 V_{RHE}$) would result in noticeably shorter Cu-O distances (ca. 1.836 Å) than those observed in the fitting of the EXAFS data collected for Cu-N-C catalysts under pulsed reaction conditions.”*

3. The rationale behind selecting specific cathode and anode potentials to control catalyst size presents an intriguing aspect of the study. Providing insights into the optimization of these potentials would enhance the manuscript's completeness and relevance.

Our response: We fully agree with the Reviewer that the choice of the cathodic and anodic potentials for pulse reaction protocols plays an important role. In fact, we are of the opinion

that the systematic investigation of the roles of these parameters deserves a separate manuscript, and we are currently discussing with experts in Design of Experiment approaches possibility to try to rationally design the optimum pulse conditions for these type of experiments, including optimizing the pulse shape, anodic and cathodic potentials and applied potential times. Here we note that the cathodic potential value (E_c) value (-1.35 V) for pulsed CO₂RR was the same as that used as the working potential in the experiments under static CO₂RR conditions. The latter, in turn, was chosen to maximize the Faradaic efficiencies for CO₂RR products under static conditions (Ref.[29]). The anodic potential (E_a) value (+0.44 V_{RHE}), in turn, was chosen after several experimental trials, with the idea to ensure that the rates of Cu reduction and regeneration of cationic Cu species are comparable. For example, if a lower E_a value is chosen (e.g., -0.16 V_{RHE} (Figure R1a)) and the durations of cathodic and anodic potential pulses are the same (30 s in the example in Figure R1), the regeneration of the cationic Cu species is slow, and the catalyst quickly becomes dominated by large metallic particles. To increase the fraction of ultrasmall species or singly dispersed cationic sites in this case, one would need to reduce strongly the duration of the cathodic pulse. However, such short pulses would be challenging to track with our present experimental methods, considering that the time resolution of our QXAFS measurements is about 1 s per spectrum. On the other hand, if the anodic potential is much higher (e.g., +0.84 V (Figure R1b)), the redispersion of metallic clusters is very fast, and catalyst is dominated by the contribution of singly dispersed cationic Cu species (which favors parasitic hydrogen evolution). Furthermore, the catalyst operation at such high anodic potentials could result in reoxidation of the reaction products, which would complicate the interpretation of the selectivity trends. We have briefly summarized this point in the revised version of the manuscript.

Figure R1 (for review only): results of LCA-XANES for the Cu-N-C catalyst under pulsed CO₂RR with $\Delta t_c = \Delta t_a = 30$ s, $E_c = -1.35$ V and different E_a values: $E_a = 0.16$ V (a) or $+0.84$ V (b). Evolution of the concentration of singly-dispersed Cu sites is shown. The LCA-XANES results obtained for Cu-N-C catalyst under static CO₂RR are also shown as dashed magenta lines.

Changes to manuscript:

In the Section “Cu particles under static and pulsed CO₂RR”, we modified the text as follows “The applied potential values were chosen so that E_c is sufficiently negative for the formation of Cu clusters during the cathodic pulse, while E_a is sufficiently positive for their redispersion during the anodic pulse.²⁹ In addition, the empirically selected pulse potential values ensured comparable rates of cluster formation (under cathodic potential) and redispersion (under

anodic potential). Future work is planned taking advantage of computational design of experiment approaches to rationally select the optimum potential pulse parameters, including their shape, duration and anodic and cathodic potential values.”

4. To substantiate claims regarding the size of Cu catalysts, it is recommended to include corresponding High-Angle Annular Dark Field Transmission Electron Microscopy (HAADF-TEM) images depicting different stages of catalyst growth. This addition would bolster the clarity and credibility of the findings.

In conclusion, this work has made notable strides in unraveling the intricate dynamics of CO₂RR, with their study underscoring the importance of structural characterization in tailoring catalyst design. Addressing the aforementioned points will undoubtedly strengthen the manuscript and contribute to its significance within the field.

Our response: We thank again the Reviewer for his/her positive remarks. Regarding the TEM images, we note that HAADF-STEM images of the as-prepared catalysts and catalysts after static and pulsed CO₂RR are already provided in Supplementary Figure 1. We had already considered the valuable suggestion of the reviewer before submitting the paper, but we decided against conducting and displaying such data since in the present case, since ex-situ/post-mortem measurements cannot provide any valuable insight into the sizes of the clusters and particles present during the catalyst operation. As demonstrated in our current work, our previous study [Hursán, D. et al. *Advanced Materials* 36, 2307809 (2023) (Ref.[29] in this manuscript)], and numerous previous literature reports (Refs.[12-19] in the manuscript), the unique property of the Cu-N-C catalysts is that the formation of metallic clusters is completely reversible. The particles formed under reducing conditions are quickly redispersed, once the reducing conditions (applied potential in the CO₂RR environment) are lifted. Thus, no correlation between the particle sizes in post-mortem analysis and the particle sizes under working conditions can be established. Moreover, providing such information in the present manuscript would be misleading to the readers, since they could think that the changes observed in the structures and sizes in TEM after the different operation cycles are irreversible, and thus the post-mortem analysis is representative of the real working catalyst structure, which is certainly not the case here. We consider that it is the reversible nature of such structural transformations which makes mandatory spectroscopy operando characterization work such as the one we are displaying here.

Reviewer #2:

In this work, the authors applied pulsed CO₂RR protocol and operando quick XAS spectroscopy to study the behaviour of single-Cu-sites catalyst. They found that the Cu particle size can be rationally tuned by the pulsed electrolysis, which was confirmed by the operando quick XAS. Combining with the CO₂RR results, they investigated the relationship between Cu size and product selectivity. Overall, the topical is interesting, well done and the manuscript is well organized. Revision is recommended.

1. Although I appreciate that the focus of this work is on the fundamental catalysis, I worry that the outcomes may not be applicable to practical systems. Specifically - could a catalyst

system that is this dynamic, be stable for 1000 hours? The < 30% loss over 5 hours is notable, but this is still a lot of catalyst loss. I don't expect the authors to generate 1000h stability, but the MS would be strengthened if this question could be addressed directly - perhaps in the discussion.

Our response: We appreciate Reviewer's comments and thank her/him for the positive comments and very helpful suggestions. As the Reviewer already pointed out, the focus of our work is indeed on establishing fundamental understanding of structure-properties relationships in Cu-N-C catalysts, particularly focusing on the challenging regime where ultrasmall metallic clusters are formed. We do not aim here to propose an economically viable catalytic system, ready for industrial applications. As suggested by the Reviewer, we have modified the "Discussion" section to make this point more clear. This being said, we also would like to point out that the catalyst loss observed in this study is comparable to what we previously observed for conventional (more bulk-like) Cu catalysts under pulsed CO₂RR conditions [Timoshenko, J. et al. Nature Catalysis 5, 259 (2022). (Ref.[20] in this manuscript)], and that we have not yet spent sufficient time trying to minimize the material loss, which might be in fact possible through a more rational approach in the selection of the reaction conditions and pulses, for instance, taking advantage of computational design of experiment approaches.

Changes to manuscript:

In the "Discussion", we have modified the text as follows *"More importantly, it provides a pathway for understanding the fundamental properties of these distinct catalytically active Cu species that we can now controllably (re-)generate through a careful selection of the dynamic reaction conditions. Thus, the main outcome of our work is the demonstration of a catalyst operation method that allows to explore structure-reactivity correlations within a large structure parameter space."*

2. The author assumed four processes during the pulse CO₂RR, (i) reduction of singly dispersed cationic Cu sites during the cathodic pulse; (ii) increase in the size of metallic Cu particles during the cathodic pulse; (iii) fragmentation of Cu particles into smaller clusters during the anodic pulse, and (iv) conversion of small Cu clusters back into singly dispersed cationic sites during the anodic pulse. Does the Cu species formed in each step, either particle or cluster or sites, possess the same local coordination structure? This is of particularly importance, as the coordination environment influences activity. To solve this question, more analysis on the XAFS data and more characterization is required.

Our response: This is a very important point raised by the reviewer. As explained in our manuscript, our analysis of EXAFS and XANES data suggests the presence of at least three different species with clearly distinct local structures: (i) cationic Cu single sites, coordinated to 4 nitrogen atoms, (ii) ultrasmall metallic Cu clusters, dominated by strongly undercoordinated sites, and (iii) larger metallic Cu nanoparticles where undercoordinated surface sites are supported by bulk-like Cu sites in the particle core. We completely agree with the Reviewer, and demonstrate this in our manuscript, that these species indeed exhibit very different catalytic properties. We emphasize, nonetheless, that this is the minimal model that is necessary to explain the observed trends in the XAS data. It is conceivable (and quite

likely) that in addition to ultrasmall clusters and large nanoparticles, a continuous distribution of metallic clusters/particles of intermediate sizes is present in the sample. However, more accurate quantification of this distribution is impossible due to the sample-averaging nature of the XAS method, and the fact that the spectra for these particles of intermediate sizes can be essentially expressed as a linear combinations of spectra for very small and very large particles. Nonetheless, since for Cu particles with sizes between 1 and 15 nm the dependency of the catalytic properties on particle size is relatively weak and smooth (as established in [Reske et al, J. Am. Chem. Soc. 136, 6978 (2014) (Ref[25] in this manuscript)]), we believe that our simplified model is still useful for explaining structure-properties relationships in the materials. Furthermore, our results suggest that even among singly-dispersed cationic Cu sites there could be variations in their local structure, as evidenced by the different degree of their reducibility, maybe even related to their specific location on the N-doped carbon matrix, which is also inhomogeneous and expected to experience modifications during the pulse operation. However, the spectroscopic differences between these species are minor (as demonstrated by the absence of additional species in our principal component analysis), suggesting that their local structure is not too different from that of the average Cu-N₄ structural motif.

3. More characterization is needed for the Cu samples post pulse electrolysis, such as HRTEM, XRD, which will give direct information on the particle size.

Our response: HAADF-STEM images of the as-prepared catalysts and the catalysts after static and pulsed CO₂RR are already provided in Supplementary Figure 1. At the same time, we believe that these (or any other) ex-situ/post-mortem measurements cannot provide any valuable insight into the sizes of the particles present during the catalyst operation, considering that the formation of metallic clusters in Cu-N-C catalysts is completely reversible, and that the particles formed under reducing conditions are quickly redispersed, once the reducing conditions are lifted, in particular, once the applied potential is removed prior to the ex-situ post-mortem analysis suggested. Thus, no correlation between the particle sizes in post-mortem analysis and the particle sizes under working conditions can be established. We furthermore note that even the largest particles observed in our study (ca 1-2 nm in size) are too small and have too disordered structures to be reliably analyzed by XRD. Moreover, providing such post-mortem information in the present manuscript would be misleading to the readers, since they could think that the changes observed in the structures and sizes in TEM after the different operation cycles are irreversible, which is certainly not the case here. We consider that it is the reversible nature of such structural transformations which makes mandatory spectroscopy operando characterization work such as the one we are displaying here.

4. It is hard to believe that, when the single-Cu-site regenerated under anodic potential, namely "mobile ultradispersed species", it can remain mobility on the electrode surface, yet not be dissolved into the electrolyte, especially considering that Cu is readily corroded under an anodic potential. This needs quantitative and direct support.

Our response: We agree with the reviewer that this is a very important aspect to be considered during pulse operation, since even for much larger Cu particles we have previously observed

such dissolution for high oxidative anodic potentials [Timoshenko, J. et al. Nature Catalysis 5, 259 (2022) (Ref.[20] in the manuscript)]. Nonetheless, the quantitative assessment of catalyst dissolution (or lack of thereof) for the present material was already given in our Supplementary Figure 8. As one can see, both, under static and pulsed CO₂RR conditions the catalyst remains remarkably stable, and the catalyst loss is less than 20% in 5000 s of experiment. This stability of the Cu species can be explained by the strong interactions of the Cu species with the N-C support, which also ensures the reversibility of Cu cluster formation, and prevents the metallic particles from unlimited agglomeration. We have added this point to the revised version of the Supplementary Information. Moreover, we note that even for the conventional (bulk-like) Cu catalysts without such particle-support interactions, the dissolution/corrosion under pulsed CO₂RR is rather limited (see Ref.[20]), even at anodic potentials of +0.6 V, which is higher than +0.44 V which is used in our current work.

Changes to manuscript:

At the end of “Supplementary Note 1”, we have added: *“This stability of the Cu species can be explained by the strong interactions of the Cu species with the N-C support, which also ensures the reversibility of Cu cluster formation, and prevents the metallic particles from unlimited agglomeration”*

5. The authors obtained slightly larger Cu-Cu CN of " $\Delta t_c = \Delta t_a = 30s$ " sample than that of "static" sample. However, the CO₂RR shows a big difference between " $\Delta t_c = 150s$ " and "static" sample, especially if the C₂/C₁ ratio is calculated. Explain and support.

Our response: Our results, summarized in Figure 4b, suggest that the catalytic properties for Cu-N-C catalysts under pulsed CO₂RR with cathodic pulse durations $\Delta t_c = 30$ s and larger are similar. In particular, no significant changes are observed in the relative selectivities of the main gaseous products for $\Delta t_c = 30$ s and longer. This agrees well with our model suggesting that in this regime ($\Delta t_c = 30$ s and longer) the catalyst is dominated by the contribution of “large” nanoparticles, whose size does not change noticeably with the duration of the potential pulse. Nonetheless, there is indeed some difference between the catalytic results obtained for pulsed CO₂RR with very long cathodic pulse durations, and the results obtained under static CO₂RR. This can be attributed to particle size effects, but also to parallel irreversible changes in the catalyst, including the support structure.

Regarding the particle size effect, as mentioned in our manuscript, the difference in the Cu-Cu coordination numbers (8.5 for Cu clusters formed in the Cu-N-C catalyst under static CO₂RR vs. 10.1 for “large” Cu particles formed under pulsed CO₂RR with long pulse durations), in fact, signifies that the particle sizes in the latter case are nearly twice as large as under static CO₂RR (ca. 2.5 nm under pulsed CO₂RR vs ca. 1.3 nm under static CO₂RR). This stems from the fact that for larger particle the particle-averaged EXAFS signal is dominated by the contribution of fully coordinated Cu atoms in the bulk of the particle, and the average Cu-Cu coordination numbers change weakly upon further increase in particle size. Thus, the particle sizes observed in the potentiostatic case and under pulsed CO₂RR with long cathodic pulse durations are substantially different, and this could contribute to the observed difference in the catalytic properties. We have already mentioned this in the Section “Size dependency of Cu particles on pulse parameters”. More importantly, the effect of

potential pulses on the CO₂RR selectivity cannot be attributed solely to the changes in particle sizes. Other effects, such as irreversible degradation of the carbon support, changes in the local pH and local electrolyte composition will also contribute to the modification of catalytic properties. In particular, the roles of irreversible transformations in the catalyst and the effect of sample history are already discussed in details in Supplementary Note 6. We have also modified the manuscript text to clarify the roles of other effects on pulsed CO₂RR.

Changes to manuscript:

To the “Discussion” section, we have added: *“One should, nonetheless, acknowledge that in addition to the dynamic changes in particle size and overall structure, other effects taking place under pulsed CO₂RR conditions could also have an impact on the observed catalytic trends, such as irreversible changes in the carbon support, periodic changes in the adsorbate structures and coverage, and variations in the local pH and concentration of reactants and intermediates.”*

and a new reference [Casebolt et al, Joule 5, 1987-2026 (2021), new (ref.[34])]

6. Detailed fitting parameters should be given for all the EXAFS data fitting, including the R-factor, Debye-Waller factor, and so on. Furthermore, if we simply compare the Figure S2 and S3, it seems the k-space data used in the data fitting has better signal-noise ratio. Did the author redo the spline? More details should be provided in the method part.

Our response: We thank the Reviewer for this valuable comment. We have now included the numerical results of EXAFS data fits in the new Supplementary Tables 1-6. Regarding the difference between Figure S2 and Figure S3, we note that Figure S2 shows the raw EXAFS data (“k-space” data), while Figure S3 shows the data already after Fourier filtering (“q-space” data). As a result, the contribution of distant coordination shells and high frequency experimental noise are not present in Figure S3. This difference is already emphasized in the caption to Figure S3, as well as in the vertical axis labels.

Changes to manuscript:

We have added new Supplementary Tables 1-6, summarizing EXAFS fitting results for Cu-N-C catalysts under static and pulsed reaction conditions.

7. 6. Comment - I appreciated the synthesis re. single-atom Cu (not good at CO₂R!), vs. small clusters, vs. larger. This is useful and I agree.

Overall, a paper of interest, worthy of revision.

Our response: We than the Reviewer for his/her positive remarks and the time taken for the careful evaluation of our work that has resulted in a clearer revised manuscript.

Reviewer #3:

Our response: We are grateful to the Reviewer his/her time and thorough reading of our manuscript

Reviewer #4:

Single site metal catalysts have been a hot topic in electrocatalysis due to their unique properties of catalytic activity and selectivity. In CO₂ electroreduction, many interesting results have been obtained via single site catalysts of Ni, Co, Fe and especially Cu.

In the literature, it has been discovered that Cu was not stayed as single sites during CO₂ electroreduction, it was reduced and formed nanocluster or particle during electrolysis. However, there are still many unclear points about this process. Cuenya and her group have further studied on this Cu single site catalyst and their nanoparticle formation. By varying the durations of the cathodic and anodic pulses, and by tracking the catalyst's evolution using operando quick X-ray absorption fine structure spectroscopy, they were able to control the average size of the Cu particles with subnanometer accuracy. In particular, they also found that, singly dispersed cationic Cu species were shown to favor hydrogen production, ultrasmall Cu clusters yielded methane, while larger Cu nanoparticles - CO and multicarbon products.

This manuscript should be considered for a publication on Nature Comm after a revision.

1. the author should be more clear about the function of Zn in the materials since Zn is also well known as a selective catalyst for CO₂ reduction to CO.

Our response: We agree with the Reviewer that the role of remaining Zn species in ZIF-8-derived catalysts is often overlooked in the literature, and should be carefully considered. To this end, in our previous very recent publication on these samples [Hursán, D. et al. *Advanced Materials* 36, 2307809 (2023) (Ref.[29] in this manuscript)], we compared the catalytic properties of Cu-N-C catalysts (among other catalysts) with those of “bare” N-C catalysts which feature only the Zn species that are not completely removed during the catalyst synthesis. We observed that the latter catalyst indeed also exhibits activity to CO₂RR, which we associate both with remaining Zn species, but also N species present in all N-C catalysts. CO and H₂ were the main reaction products in this case. However, the catalytic currents of the “bare” N-C catalyst were 3.5 times lower than those observed for Cu-N-C catalyst. Thus, we conclude that the reaction products observed in our current study, stem mostly from the presence of Cu species. We have emphasized this point better in the revised version of our manuscript.

Changes to manuscript:

Section “Catalytic selectivity:” we added “*The activity of Cu-N-C catalyst was several times higher than that of the “bare” N-C catalyst, which features N and residual Zn species, remaining from the synthesis.*”²⁹

2. The formation of nanoparticle by electrocatalysis of CO₂, this nanoparticle is only Cu or bimetallic CuZn. It need to be studied more and make it clear.

Our response: This is an important point that we have now clarified in the revised manuscript. The formation of CuZn alloy can be easily identified by EXAFS analysis, since the Cu-metal distances in CuZn alloys are longer than in pure metallic Cu [Timoshenko, J. & Roldan Cuenya, B., Chem. Rev. 121, 882–961 (2021) (Ref.[48] in the manuscript)], and [Timoshenko et al, Chem. Sci. 11 2020, 3727]). The absence of such elongated Cu-metal distances in our EXAFS results allows us to rule out significant alloying of Cu with Zn in our catalyst. We have emphasized this point in the revised version of the Supplementary Information.

Changes to manuscript:

In the Supplementary Note 1, we added *“The analysis of Cu-metal interatomic distances also allows us to rule out alloying of Cu species with Zn species remaining in the Cu-N-C catalyst after synthesis from ZIF-8 precursor. Indeed, in the latter case the Cu-metal distances are expected to be noticeably longer than in pure metallic Cu.”*

3. Via XANES analysis, it is clear that Cu-N has been transformed into Cu-Cu during catalysis. Is there any existence of Cu in the reduction process?

Our response: In our manuscript, the species present in the working Cu-N-C catalysts are identified based on both, XANES and EXAFS analysis. Both methods independently show the reduction of cationic Cu species under static CO₂RR, and the formation of metallic Cu particles. As we have also emphasized in our reply to Reviewer #1, the absence of any additional species in our samples follows also from the principal component analysis of XANES spectra, which shows unambiguously that only two spectroscopically distinct species are present in our Cu-N-C catalysts (i.e., the original Cu²⁺-N₄ species, and the metallic Cu species formed under reducing conditions.)

Our response:

We have expanded “Data availability” section. We also include “Source Data” file (Excel file “Source Data.xlsx”) containing the numerical values for the data shown in Figure2a-f and Figure3a-c and the corresponding uncertainties.

REVIEWERS' COMMENTS

Reviewer #1 (Remarks to the Author):

The authors has successfully addressed all the issues raised by the reviewer. The manuscript is in good shape, and the reviewer recommend it to be accepted for publication.

Reviewer #2 (Remarks to the Author):

I am satisfied with the revision and recommend publication. I would encourage the authors to be clear in the discussion of the 5000s stability result that this level of stability - while a useful data point here - is not considered stable from an application perspective. In otherwise, the outcome of the experiment is useful for supporting the manuscript, but be clear that this is well short of what would be required in an application.

Reviewer #3 (Remarks to the Author):

Reviewer #4 (Remarks to the Author):

After considering the revised version, I agree to accept this manuscript for publication

Point-by-point response to the Reviewers' comments

The Reviewers' comments are always followed by our responses. **The action taken is highlighted in yellow.**

Reviewer #1:

The authors has successfully addressed all the issues raised by the reviewer. The manuscript is in good shape, and the reviewer recommend it to be accepted for publication.

Our response: We thank the Reviewer for his/her positive comments.

Reviewer #2:

I am satisfied with the revision and recommend publication. I would encourage the authors to be clear in the discussion of the 5000s stability result that this level of stability - while a useful data point here - is not considered stable from an application perspective. In otherwise, the outcome of the experiment is useful for supporting the manuscript, but be clear that this is well short of what would be required in an application.

Our response: We agree with the Reviewer and we have modified the corresponding paragraph in Supplementary Note 1 accordingly.

Changes to manuscript:

At the end of "Supplementary Note 1", we have added following sentence: ***"We acknowledge, nonetheless that the demonstrated stability of our catalyst on the time scale of 5000 s does not guarantee its stability on longer time scales relevant for practical application of this catalytic system, and that this deserves separate future investigations."***

Reviewer #3:

Our response: We are grateful to the Reviewer his/her time and thorough reading of our manuscript

Reviewer #4:

After considering the revised version, I agree to accept this manuscript for publication

Our response: We thank the Reviewer for his comments and positive feedback.

Additional changes:

- Suggested editorial changes have been implemented (see the Author's Checklist attached)
- We have also slightly revised the schematical depictions of single catalyst structure in Figure 3 and 4 for clarity